# Nutraceuticals Synergistically Promote Osteogenesis in Cultured 7F2 Osteoblasts and Mitigate Inhibition of Differentiation and Maturation in Simulated Microgravity

**DOI:** 10.3390/ijms23010136

**Published:** 2021-12-23

**Authors:** Justin Braveboy-Wagner, Yoav Sharoni, Peter I. Lelkes

**Affiliations:** 1Department of Bioengineering, College of Engineering, Temple University, Philadelphia, PA 19122, USA; jsbw@temple.edu; 2Department of Clinical Biochemistry and Pharmacology, Faculty of Health Sciences, Ben-Gurion University of the Negev, P.O. Box 653, Beer-Sheva 8410501, Israel; yoav@bgu.ac.il

**Keywords:** bone, random positioning machine, simulated microgravity, polyphenol, phytonutrient, zinc, mineralization, synergy

## Abstract

Microgravity is known to impact bone health, similar to mechanical unloading on Earth. In the absence of countermeasures, bone formation and mineral deposition are strongly inhibited in Space. There is an unmet need to identify nutritional countermeasures. Curcumin and carnosic acid are phytonutrients with anticancer, anti-inflammatory, and antioxidative effects and may exhibit osteogenic properties. Zinc is a trace element essential for bone formation. We hypothesized that these nutraceuticals could counteract the microgravity-induced inhibition of osteogenic differentiation and function. To test this hypothesis, we cultured 7F2 murine osteoblasts in simulated microgravity (SMG) in a Random Positioning Machine in the presence and absence of curcumin, carnosic acid, and zinc and evaluated cell proliferation, function, and differentiation. SMG enhanced cell proliferation in osteogenic medium. The nutraceuticals partially reversed the inhibitory effects of SMG on alkaline phosphatase (ALP) activity and did not alter the SMG-induced reduction in the expression of osteogenic marker genes in osteogenic medium, while they promoted osteoblast proliferation and ALP activity in the absence of traditional osteogenic media. We further observed a synergistic effect of the intermix of the phytonutrients on ALP activity. Intermixes of phytonutrients may serve as convenient and effective nutritional countermeasures against bone loss in space.

## 1. Introduction

Long-term space missions, like the 3–6-month long sojourn of astronauts onboard the ISS, may harmfully impact the health of human space travelers. Microgravity has been shown to adversely affect the human body, especially bone tissue, impacting skeletal health [1,2]. Prior studies have suggested that microgravity imbalances the equilibrium of osteogenesis and bone resorption, mediated by osteoblasts and osteoclasts, respectively [3]. Although the rate of bone loss varies depending on the anatomical location, the net result is substantial bone loss, particularly in load-bearing bones [4,5]. While experiments in orbit are ideal for microgravity research, similar results can be obtained on Earth using the simulated microgravity (SMG) environments provided by clinostats like the Rotating Wall Vessel (RWV) Bioreactors or the Random Positioning Machine (RPM). Ground-based microgravity simulation is an attractive alternative given the expense and difficulty of space-based research [6,7].

One of the mechanisms by which microgravity can affect tissue function is by promoting the generation of reactive oxygen species (ROS), including free radicals like peroxide and superoxide [8]. Past studies have found elevated levels of oxidative stress during space flight, as microgravity conditions create inflammatory-like responses and lead to the activation of specific stress responses [9]. Conventional dietary antioxidants, such as vitamins E and C, have previously been shown to mitigate oxidative damage caused by free-radicals [10]. Recently, Morabito et al. observed that a dedicated antioxidant, Trolox, counteracted SMG-induced damage in MC3T3 osteoblasts, restoring intracellular ROS and Ca^2+^ levels, mitochondrial membrane potential, and actin filament length but not nuclear circularity [11]. Nutraceuticals, such as the phytochemicals curcumin and carnosic acid, or trace elements, like zinc, also have significant antioxidant properties and may thus be efficient in mitigating some of the adverse effects of space flight [12,13].

Curcumin (diferuloylmethane or (1E,6E)-1,7-bis(4-hydroxy-3-methoxyphenyl)-1,6-heptadiene-3,5-dione) is a phenolic compound isolated from the rhizomes of the turmeric plant, *Curcuma longa*. Historically, it has been used as a dietary spice, particularly in Asia. It has been used in traditional Indian medicine to treat inflammation, hepatic disorders, and sinusitis [14]. Curcumin has anti-inflammatory and antioxidant properties [15], including regulation of endoplasmic reticulum (ER) stress [16] and inhibition of diverse transcription factors, such as nuclear factor κB (NF-κB) in cancer cells [17,18]. Curcumin affects osteoblast differentiation in mesenchymal stem cells by increasing the expression of genes related to osteogenesis, like runt-related transcription factor 2 (Runx2) and alkaline phosphatase (ALPl), and by increasing ALP enzymatic activity and mineralization through the mediation of ER-stress [19]. Although the hydrophobicity of curcumin limits its activity by reducing bioavailability, phytochemical plant extracts like curcumin are attractive targets for inclusion in regenerative medicine although to date, only a few attempts have been reported in bone [20].

Carnosic acid (CA), like curcumin, is a plant-derived phenolic (catecholic) diterpene with antioxidative and antimicrobial properties. It is extracted from sage (*Salvia carnosa*) or rosemary (*Rosmarinus officinalis*) and utilized commercially in the food, nutritional health, and cosmetics industries [21]. Carnosic acid and an oxidized derivative, carnosol, protect lipids from oxidation, and function as a quencher for reactive oxygen species (ROS) [22]. In addition to decreasing the expression of molecules contributing to ROS, CA inhibits osteoclastic markers in bone and suppresses the Receptor Activator for Nuclear Factor-κB Ligand (RANKL) activity in vitro [23]. Furthermore, CA inhibited osteoclastogenesis and bone resorption and protected against joint destruction in vivo [24]. It is believed that CA shows some promise as a potential in-vivo osteogenic agent primarily due to its known action on osteoclasts, while in osteoblasts, it has proven to be an antioxidant, significantly attenuating H_2_O_2_ (ROS) levels in MC3T3 cells in a concentration range of 1 to 10 μM but also decreasing day-10 ALP activity and day-12 mineralization at a higher concentration of 3 to 10 μM [25].

Zinc, a trace element that is a critical catalyst in numerous enzymes and proteins, is essential for health and regulatory functions in the body. Zinc deficiency results in the retardation of bone growth [26]. Osteoporosis in human patients [27,28] and in aging rats [29] is accompanied by decreased levels of skeletal zinc, while zinc supplements have preventive and therapeutic effects, counteracting bone loss [30]. Skeletal unloading, a ground-based model for microgravity, also affects the levels of zinc in rats [31] and humans [32].

As skeletal unloading and osteoporosis have similar effects on bone mass loss as gravitational unloading [33,34] we hypothesized that curcumin and carnosic acid, along with zinc, may provide nutritional countermeasures against the detrimental effects of reduced gravity. In testing our hypothesis, we aimed to characterize the effects of curcumin, carnosic acid, and zinc on osteoblast differentiation in murine osteoblasts using the 7F2 cell line cultured in SMG (~10^−3^ × G) in the Random Position Machine. The effects of these three nutritional supplements on gene expression, enzymatic activity, and long-term mineralization in osteoblastic cells in SMG have not yet been investigated. In this study, we focused primarily on the effects of the nutraceuticals on the expression of three osteogenic marker genes: Runx2, Osteonectin (ON), and alkaline phosphatase (gene) (ALPl). Furthermore, we investigated the effects of the nutritional supplements on the ALP enzymatic activity and long-term mineralization.

## 2. Results

To determine the effects of nutraceuticals on 7F2 osteoblasts on Earth (1G control) and in SMG (~10^−3^ × G), the cells were first grown to confluence under 1G conditions in maintenance media (MM). Confluent monolayers were then exposed to SMG (experimental) or 1G normal (control) conditions and cultured for six additional days in keeping with previous experiments that identified a peak of gene expression and ALP activity at day 6 under these experimental conditions [35]. In some experiments, assessing the effects of long-term culture, the cells were maintained for 21 days after being transferred into SMG.

### 2.1. Nutraceuticals Affect Cell Viability and Increase Cell Numbers:

Previous studies have explored the potential cytotoxicity of curcumin [19], carnosic acid [36], and zinc [37] for osteogenic cells, using several different viability assays. A fluorescent live/dead stain (calcein-AM ethidium homodimer-1, see Methods) was used to experimentally evaluate the effects of these nutraceuticals on the initial viability for 7F2 cells. In preliminary studies, and in line with prior results in other cell lines [16,38,39,40], we established that all nutraceuticals reduced short-term cell viability by ≤25% (Table 1), a “shock effect” distinct from longer-term effects on cell proliferation. A similar effect has been described before in the literature for low levels of carnosic acid (6 µg/mL) in various cell lines [36].

To assess the effects of SMG on cell numbers in differentiating 7F2 monolayers, all samples were initially seeded at the same density in MM, brought to similar levels of confluence, and then cultured in DM for another six days, either at 1G or in SMG in the presence or absence of the nutraceuticals. In these longer-term cultures, reflecting the subsequent enzymatic and gene expression studies (six days, see below), the presence of the nutraceuticals had a biphasic effect on cell numbers in 1G, with cell counts peaking at low concentrations (2.5 μM curcumin, 5.0 μM carnosic acid, 50 μM zinc), then diminishing with increasing concentrations (Appendix A). By contrast, in SMG, low concentrations of nutraceuticals did not affect cells numbers. Hence, the maximal “safe” concentrations used for subsequent studies (both short- and long-term) were 5 μM for curcumin, 10 μM for carnosic acid, and 50 μM for zinc.

As seen in Figure 1, in the absence of nutraceuticals, the cell numbers in SMG increased by ~190% over the 1G controls. Furthermore, also in 1G, the addition of nutraceuticals led to an increase in the cell numbers by ~145% for the concentrations of the phytonutrients listed and an approximate doubling of cell numbers for zinc. In SMG, the nutraceuticals did not cause a significant increase cell numbers, in contrast to their effect in 1G.

### 2.2. Nutraceuticals Modulate Short-Term ALP Activity

Alkaline phosphatase is an established primary marker for osteoblast maturation and mineralization [41,42]. Here, we tested the effects of the nutraceuticals on ALP activity in 7F2 cells cultured in osteogenic medium in both 1G and SMG following six days’ culture in the absence and presence of various concentrations of nutraceuticals (for details see the Methods section). As seen in Figure 2, the culture of control/untreated cells in SMG reduced ALP activity by 42 ± 9%, similar to the reduction seen in our previous studies [35].

Under 1G conditions, all three nutraceuticals tested caused significant increases in enzymatic ALP when the cells were cultured in osteogenic medium. For example, in the presence of curcumin (7.5 μM), ALP activity increased on average by 38 ± 6%, while it increased by 59 ± 16% for carnosic acid (10 μM) and 61 ± 27% for zinc (50 μM). (Figure 2).

In SMG, this increase was even more pronounced: ALP activity was elevated 140± 13% for (7.5 μM) curcumin, 113 ± 25% for (10 μM) carnosic acid, and 160 ± 20% for (50 μM) zinc versus the untreated SMG controls at these same concentrations. Thus, the nutraceuticals tested essentially abrogated the inhibitory effects of SMG. The reversal of the SMG-induced inhibition of ALP activity by the nutraceuticals was dose dependent, yielding increases of 23%, 25%, and 35% at the medium concentrations (5.0 μM curcumin, 10 μM carnosic acid, and 100 μM zinc) and 40%, 81%, and 30% at the maximal concentrations (7.5 μM Cur, 25 μM CA, and 250 μM Zn) (Figure 2). Unlike the two phytonutrients, zinc produced the strongest increase in ALP activity at the lowest concentration of 50 μM, with increasing dosages having no significant additional effect in 1G and decreasing effectiveness in SMG.

### 2.3. Nutraceuticals Induce Osteogenic Marker Gene Expression in Non-Osteogenic Medium

The expression and upregulation of osteogenic marker genes, together with the increase in ALP activity, confirm osteogenic differentiation. The effect of the nutraceuticals on the mRNA expression of three distinct osteogenic genes (ALPl, Runx2, and ON) was determined using quantitative RT-PCR. Following six days of culture in osteogenic differentiation media, osteogenic markers were studied in both 1G and SMG (Figure 3).

In line with our previous results [35], SMG, in general, suppressed the expression of all three genes tested by approximately 60%. Surprisingly, while the nutraceuticals did not significantly alter gene expression in osteogenic differentiation media, the nutraceuticals did significantly induce the expression of all osteogenic genes in non-osteogenic maintenance media (Figure 3). As an example, in SMG, in the presence of all nutraceuticals, ALPl expression in non-osteogenic maintenance media (SMG MM) was elevated to levels statistically similar to that seen in SMG osteogenic media (SMG DM). This contrasts with osteogenic differentiation media (DM), where the nutraceuticals did not change the expression of ALPI, Runx2, or ON. While the nutraceuticals did not effectively counteract the inhibition of osteogenic marker gene expression in SMG, supplementation with the nutraceuticals raised the expression levels of 1G MM to 1G DM control and SMG MM to SMG DM control. Using ALPI as an example, the increases observed for cells cultured in MM-media were 160% for curcumin, 130% for carnosic acid, and 170% for zinc, respectively.

### 2.4. Phytonutrients Display Synergistic Effects in Intermixtures Independent of Zinc

To explore the previously described synergistic effects of adding diverse phytonutrients together, we evaluated ALP activity in the osteogenic monolayers (cultured in DM) in the presence of a combination of the nutraceuticals, specifically when the phytonutrients curcumin and carnosic acid were tested together (“intermix”). When curcumin and carnosic acid were mixed (3.75 µM curcumin + 12.5 µM carnosic acid), the resulting ALP activities were significantly greater than the effects of the individual nutraceuticals when factored additively (Figure 4), suggesting synergy between these two phytonutrients.

The intermix of curcumin and carnosic acid increased ALP activity approximately two-fold, 113 ± 37% in 1G and by 90 ± 33% in SMG, compared to the hypothetical additive of curcumin and carnosic acid individually. By contrast, when either phytonutrient was mixed with zinc (3.75 µM curcumin + 25 µM zinc or 12.5 µM carnosic acid + 25 µM zinc) additive effects were observed, where the total ALP activity was statistically not different from the (theoretical) sum of the individual nutraceuticals in both in 1G and SMG. The triple intermix of all three nutraceuticals also displayed a synergistic reaction, especially in SMG, due to the presence of the phytonutrients within the mixture.

### 2.5. Long-Term Application of Nutraceuticals Partly Mitigated Inhibition of ALP Activity and Mineralization by SMG

To simulate the potential effects of nutraceuticals on the long-term osteogenic behavior of mature mineralizing osteoblasts, 7F2 cells were plated as before in 1G MM, grown for about six days to near confluence (see Methods), and then cultured for another 21 days (instead of six days) in DM under SMG conditions or in 1G. Experiments were performed in osteogenic media and nutraceuticals were replenished during periodic media changes following a cycle of two days, two days, and three days.

We focused on assessing long-term ALP activity, as it persists during the mature mineralizing phase and is essential in the production of hydroxyapatite. As seen in Figure 5, SMG strongly inhibited ALP activity under nutraceutical-free conditions, reducing activity to 28 ± 7% (*p* < 0.0001) of 1G control.

Nutraceuticals counteracted this inhibitory effect to differing degrees, resulting in 40 ± 12% of 1G control (*p* < 0.0001) in the presence of zinc, and 61 ± 24% of 1G control (*p* < 0.0001) for the phytonutrient mix (Figure 5a). Expression relative to 1G nutraceutical was 68 ± 15% for zinc (*p* < 0.05) and 108 ± 29% for the intermix (not significantly different). Mineralization was analyzed in parallel to determine amounts of calcium deposited on the flask surface area after 21 days. SMG reduced mineralization to 70 ± 6% of 1G control. Zinc significantly (*p* < 0.05) recovered mineralization in SMG, resulting in 82.5 ± 15% of 1G control (Figure 5b). Interestingly, though the phytonutrient intermix increased ALP activity more than zinc (Figure 5a), it did not significantly (*p* = 0.9985) increase mineralization.

## 3. Discussion

Reductions in bone mass and loss of skeletal mineral, particularly in load-bearing bones, remain a serious health complication associated with space travel. Microarchitectural deterioration occurs due to increased resorption and decreased mineralization, resulting in lower bone mass and leading to an elevated risk of fracture [3]. There is an urgent, unmet need for (nutritional) countermeasures that could mitigate the detrimental effects of space flight on bone health. In this study, we demonstrated that the polyphenol derivatives curcumin and carnosic acid and the trace element zinc can promote osteoblastic cell growth over six days and, importantly, that all three nutraceuticals stimulated osteogenic differentiation in the absence of traditional osteogenic media, as assessed by elevated enzymatic ALP activity (Figure 2) and changes in the expression of osteogenic marker genes (Figure 3). The osteogenic effects of the plant-derived nutraceuticals were identified as synergistic (Figure 4). Cells treated with the nutraceuticals were able to counteract inhibition due to SMG by increasing the enzymatic activity of tissue-nonspecific alkaline phosphatase without elevating its mRNA expression levels [43].

Plant phytochemical derivatives, such as curcumin or carnosic acid, are well-known nutritional supplements with anti-oxidative properties and have been widely studied, e.g., for use as anti-cancer agents. The beneficial effects of these nutraceuticals on bone health are believed to stem from either a protective effect against osteoclastic bone resorption or a direct augmenting effect on osteoblastic bone formation [24,25,36,39]. Carnosic acid and phenolic extracts are known to have anti-microbial properties [44] and inhibit inflammation [24]. CA acts as a protective compound by activating signaling pathways associated with cell survival [45] by promoting mitochondrial protection [46,47]. To the best of our knowledge, there are few, if any, studies that have been published on osteoblasts and carnosic acid. By contrast, curcumin has shown osteogenic effects in prior studies [19].

Zinc is likewise known to be essential to bone health and to have osteogenic properties [37,48,49,50] but is structurally and functionally distinct from phytonutrients derived from plants. Zinc has been shown to have anabolic effects on osteoblastic MC3T3-E1 cells, stimulating protein and DNA synthesis [30,51,52], while zinc deficiency can inhibit osteogenic gene expression [53]. Zinc increases alkaline phosphatase activity [48,54] in osteoblastic MC3T3-E1 cells and enhances collagen synthesis during bone formation [51]. It also stimulates osteoblastic bone formation [50,52]. The consensus in the literature is that zinc likely plays an important role in the signaling pathways of osteogenesis and that dietary zinc supplements can alleviate the deleterious effects of skeletal unloading [31,55] and osteoporosis [27].

In this study, we used the murine osteoblastic cell line 7F2 to test the hypothesis that antioxidant nutraceuticals, specifically the phytonutrients curcumin and carnosic acid, might promote osteoblast differentiation and function and thus potentially mitigate inhibition of osteogenic differentiation in microgravity. We used zinc as a positive osteogenic control, one that would operate distinctly from the phytonutrients.

In this paper, we observed a short-term “shock effect”: exposure for 6 h to the nutraceuticals impacted cell viability in a dose-dependent fashion (Table 1).Wee This finding is in line with some prior publications. For example, previous studies have shown inhibition of proliferation of rat calvarial osteoblasts (ROB cells) by curcumin [20,56]. Pesakhov et al. observed a 10% drop in peripheral blood mononuclear cell viability by 5 µM curcumin and 10 µM carnosic acid from 0 to 72 h; while combinations of the two phytochemicals at these concentrations did not significantly reduce the viability of peripheral blood monocytes (PBMCs), they reduced the viability of HL-60 cells by more than 40% [39]. However, in the longer-term (six days) studies (Figure 1), exposure to the “safe doses” of nutraceuticals resulted in an increase in the cell numbers. Taken together, these studies indicate that the cytotoxicity of these phytochemicals can be cell-type specific and needs to be established on a case-by-case basis [40].

Proliferation, as observed in this study, is quite different from the conventional cytotoxicity/proliferation data from most other studies in that our cell cultures prior to exposure to nutraceuticals were already confluent. Thus, exposure to either osteogenic media or the nutraceuticals themselves pushed the cells from a quasi-quiescent /confluent phenotype into a differentiating/proliferative one. Untreated control cell populations showed no significant change from confluence, whereas in nutraceutical-treated samples, the cell numbers per unit area were elevated in both 1G and SMG (Figure 1). Preliminary studies suggest that the transfer of the confluent monolayers from 1G to SMG caused a significant decrease in cell size/ surfaced area, thus permitting a more tightly packed monolayer [57]. Antioxidants have been shown to restore some proliferation inhibition due to SMG and, in this same study, counteract some changes in cell morphology (filament length) though not in others (circularity) [11]. We speculate that the increase in cell numbers upon exposure to DM and SMG may be related to this decrease in cell surface area [57].

Compared to the copious literature describing the cytotoxic effects of curcumin and its use as an anti-cancer agent [36,58,59], there is less information available about the effects of curcumin on osteoblasts. There are divergent opinions/findings of curcumin either inhibiting or promoting osteogenesis. For example, curcumin has been shown to decrease mineralization in rat calvarial osteoblastic cells out to 14 days in a time-dependent manner [56]. In line with this observation, we noted a consistent decrease in long-term 1G mineralization and ALP activity (Figure 5).

On the other hand, there is precedent for nutraceuticals like curcumin being osteoinductive, even in the absence of osteogenic media [20]. In line with this notion, in our short-term studies (six days), we observed significantly elevated levels of ALP (Figure 2) in both osteogenic media and non-osteogenic media as well as the transient activation of gene markers for differentiation in nutraceutical-supplemented samples (Figure 3), which is not sustained over time and with the maturation of the osteoblasts. Due to its antioxidant and anti-inflammatory properties, curcumin has also been previously investigated as a component of biomaterials and scaffolds for regenerative medicine and bone repair [20].

Alkaline phosphatase is an essential component for the formation of hydroxyapatite and the maintenance and mineralization of bones; it facilitates the deposition of inorganic phosphate (Pi) by catalyzing the conversion of pyrophosphate (PPi). Osteoblasts secrete ALP enzymes during the matrix maturation and early-mineralization phases of their life cycle although not during their proliferative phase. In osteoblastic cell lines, like 7F2s, MC3T3-E1s, either the expression/upregulation of ALP genes, or the enzymatic activity of ALP gene products serve as a strong indicator for osteogenic differentiation. ALP activity plays a key role in the mineralization and maintenance of bone. However, as demonstrated by Sugawara et al. (2002), while the enzymatic activity of ALP is necessary for mineralization in MC3T3 cells, elevated levels of expression or activity do not necessarily result in increased mineralization [42]. Our results showed that ALP expression was highest in the short-term, within six days, and that within at least this window of phytonutrient concentrations (1 µM to 5 µM curcumin, 1 µM to 10 µM carnosic acid), 7F2 differentiation was increased significantly in both osteogenic media (Figure 2) and maintenance media (Appendix A). However, this nutraceutical advantage was lost over the longer term, suggesting a time-sensitive component to exposure or a decrease in the excitatory sensitivity of mature mineralizing cells versus immature cells in the matrix maturation stage.

To the best of our knowledge, no prior studies have investigated the interaction between mechanical unloading (e.g., gravitational unloading) and the effects of these specific nutraceuticals on osteoblasts. In both short- and long-term studies, nutraceuticals counteracted the SMG-mediated inhibition of mineralization and differentiation. In the short term, after the initial six days, ALP activity was elevated to levels equal to or greater than 1G controls. In the long term, after 21 days, while at 1G, ALP activity in the nutraceutical-treated samples remained below the control, it was significantly elevated in SMG upon nutraceutical supplementation versus untreated SMG controls (Figure 5). Mineralization, while also decreased from 1G, was either partly recovered or unchanged in the samples maintained in SMG, unlike in 1G.

The lack of changes in gene expression in both treated and untreated SMG samples (Figure 3) suggests that the observed elevation of ALP activity and mineralization may be a nutri-epigenetic effect. Similar effects, i.e., enhanced mineralization and ALP activity with minimal (<50%) or no significant changes in osteogenic gene expression, have been previously reported in MC3T3-E1 osteoblasts treated with zinc and genistein [60]. Furthermore, curcumin appears to play a role as an epigenetic modulator in general [43] and specifically as an epigenetic modulator of miRNAs involved in the osteogenic differentiation of dental pulp stem cells [43,61]. Going forward, we suggest focusing on the mechanisms by which the three nutraceuticals studied can improve bone health in microgravity, for example, by altering post-translational regulation of the transcriptional activity of osteogenic genes or serving as modulators of the epigenetics of osteogenic miRNAs.

We further identified a synergistic increase in ALP activity although not gene expression in the cells concomitantly treated with both phytochemicals, curcumin and carnosic acid, and observed osteogenic differentiation stimulated by all three nutraceuticals (Figure 4). In the case of synergy under 1G conditions, the curcumin-carnosic-zinc intermix may have reached the effective ceiling of ALP activity in these cells under these conditions due to a limited supply of substrate or a maximal activity of the enzyme. Maximal ALP activity in SMG, akin to that in 1G, was obtained in the triple mix of the nutraceuticals and might reflect the above-mentioned plateau. A similar plateau of ALP activity was also found in other papers testing additive effects of nutraceutical trace elements, like Zn^2+^ and strontium [62]. Due to this plateau effect at 1G, inhibition by SMG allowed the synergistic amplification to be more acute in the SMG sample than in the 1G (Figure 4).

In our hands, zinc appeared to act on an unrelated pathway that did not synergize with the phytochemicals. However, zinc was effective in stimulating osteogenic differentiation (Appendix A) and gene expression (Figure 3) in maintenance media. Zinc was also effective in counteracting gravitational inhibition of ALP activity in the short term (Figure 2) and in partially increasing mineralization in the long term (Figure 5b).

These nutraceuticals may be useful as potential candidates for nutritional countermeasures to health challenges due to spaceflight, specifically alleviating skeletal damage caused by exposure to microgravity. As these nutraceuticals seem to at least partly mitigate the effects of gravitational unloading, future research may be warranted to investigate these compounds for osteo-assistive effects in situations involving mechanical unloading here on Earth, such as extended bed rest.

## 4. Materials and Methods

### 4.1. Materials

Alpha-Minimum Essential Medium (α-MEM) and Fetal Bovine Serum were purchased from Gibco Life Technologies (Carlsbad, CA, USA). L-ascorbic acid, β–glycerophosphate (β-GP), para-Nitrophenylphosphate (pNPP), Alizarin Red, zinc sulfate, and Tri Reagent^®^ for processing tissues were from Sigma-Aldrich (St. Louis, MO, USA). The Quant-iT™ PicoGreen™ dsDNA Assay Kit was purchased from Invitrogen Molecular Probes (Eugene, OR, USA) via Thermo Fisher Scientific. TaqMan Fast Universal PCR Master Mix (2X) and Taqman primers were from Applied Biosystems (Foster City, CA, USA). RNeasy Protect Mini Kits were acquired from Qiagen (Hilden, Germany). Crystalline curcumin (>95%) was purchased from Cayman Chemicals (Ann Arbor, MI, USA) and carnosic acid (93–97%) from Alexis Biochemicals (San Diego, CA, USA).

### 4.2. Cell Culture Techniques

7F2 murine osteoblasts (American Type Culture Collection, Manassas, VA, USA, CRL-12557) were cultured in α-MEM media supplemented with 10 mM HEPES, 10% FBS, and 1% streptomycin and penicillin and maintained in a humidified, 37°C, 5% CO_2_/air incubator (maintenance medium, MM), as previously described [35]. Cells were grown to 90% confluence in T-12.5 Falcon™ Tissue Culture Treated Flasks (Fisher Scientific, Waltham, MA, USA) in MM. For osteogenic induction, the cells were cultured in osteogenic media comprised of the above complete α-MEM medium supplemented with 10 mM β-glycerophosphate and 10 µg/mL ascorbic acid (differentiation medium, DM) and any experimental nutraceuticals. In preliminary studies, we optimized the initial seeding density to yield a robust monolayer of approximately 100k cells per flask at the time of collection (day 6 experimental). The media was refreshed by aspiration of 50% existing media and application of fresh media on the following schedule: 2 days, 2 days, and 3 days, thus generating a weekly cycle. Short-term experiments were collected after 6 days. Long-term experiments ended after 21 days.

### 4.3. Random Positioning Machine

To simulate microgravity conditions, all experiments were carried out in a Random Positioning Machine (RPM^SW^ 2.0) (DutchSpace Airbus, Leiden, Netherlands). In contrast to the 1st generation RPM, which provides random motion only, the software-driven RPM^SW^ Random Positioning Machine has two modes of operation: random and path-file. Specific path-files allow the RPM to simulate partial gravity and to reach microgravity more rapidly and reliably than the random mode [35,63,64,65]. For modeling microgravity, we used the “p0b” path file, which generates SMG of ~10^−3^ × g. To be compatible with the RPM hardware, the cells were cultured as 2D monolayers in T-12.5 Falcon™ Tissue Culture Treated Flasks, retrofitted with Fischer-brand Silicone Recessed Septum Stoppers (internal diameter: 14.5 to 15.5 mm) for enhanced gas exchange. Protocols and devices for flask-mounting, maintenance of the flasks under “zero-headspace” conditions (to minimize fluid shear stress), and bubble-free media change were as detailed previously [35]. After preliminary studies using static controls, the 1G controls were maintained on a dynamic orbital platform within the same incubator at close to 10 rpm cyclical, analogous to the motion setting (2–10 rpm) of the RPM in random mode.

### 4.4. Short Term Cytotoxicity/Cell Viability Assay

The viability of the cells in the presence of the various nutraceuticals was measured in 48-well microplates using an Infinite 200 PRO multimode plate reader (Tecan Group Ltd., Switzerland). Cells were seeded in maintenance media (see above) in 48 wells at an initial density of 20,000 cells/well. Once the cells reached 50% confluence, the old media was aspirated and replaced with the same media supplemented with test compounds (nutraceuticals) at varying concentrations. To assess the potential cytotoxicity of the compounds, cells were incubated with this media for 6 h (one-fourth of their normal replication time). Cell viability was tested using the Live/Dead Viability/Cytotoxicity Kit for mammalian cells (Molecular Probes, Eugene, OR), according to the manufacturer’s protocols. Fifteen minutes before the assay, the media of the negative controls was gently aspirated, and the cells were treated with 70% methanol in DPBS (Dulbecco’s phosphate-buffered saline). Positive controls (untreated live cells), negative controls (70% methanol treated dead cells), and background fluorescence (no cells) were all tested concomitantly with experimental wells after fifteen-minute incubation. Using the plate-reader, the wells were excited at 485 nm to visualize viable cells stained with calcein-AM and at 530 nm to visualize dead cells with stained ethidium homodimer-1 dye. Fluorescence emissions were acquired at 530 nm and 645 nm, respectively.

### 4.5. PICO Green Assay for Long-Term Cell Proliferation

The PicoGreen dsDNA Quantitation Reagent (Invitrogen, Eugene, OR, USA) was supplied as a 1-mL concentrated dye solution in anhydrous dimethylsulfoxide (DMSO) and used following the manufacturer’s protocol. In brief, a 0.1% Triton cell lysate supernatant (see below) was diluted 400x, and 100 µL of that solution was aliquoted into a 96-well plate. A total of 100 µL of the combined PicoGreen Reagent (1:200 PicoGreen diluted in the TE buffer supplied with the kit) was added to each sample. After mixing and incubation for 5 min at room temperature, protected from light, fluorescence was measured on an Infinite 200 PRO multimode plate reader (Tecan Group Ltd., Männedorf, Switzerland) at 485 nm excitation, 535 nm emission. For each experiment, standard curves were constructed based on cell lysate supernatants extracted from known cell numbers (counted in triplicate) and experimental results are expressed as cell numbers.

### 4.6. Alkaline Phosphatase Activity Assay

The enzymatic activity of ALP, a marker for osteoblastic differentiation, was quantitated spectrophotometrically, as previously described [35]. In brief, 50 k cells were seeded in T-12.5 Falcon™ Tissue Culture Treated Flasks (Fisher Scientific, Waltham, MA, USA) and grown to confluence in maintenance media. Experiments were conducted as per cell culture methods above. After either 6 or 21 days, the growth medium was aspirated, and the cells were washed (at RT) with PBS. The cell monolayers were scraped in 250 µL phosphate-buffered saline PBS and transferred into 1-mL microcentrifuge tubes. The cells were then lysed by the addition of 250 µL 0.2% Triton in PBS to a final concentration of 0.1 % (*v*/*v*) Triton X-100 (500 µL), followed by one freeze-thawing cycle (−80 °C/RT) and centrifugation (2000× *g*, 1 min). The supernatants were used to determine ALP activity according to the protocol of Lin et al. [66] with some modifications, as previously described [35]: the buffer used was 10mM MgCl_2_, 0.5M AMP (2-Amino-2-Methyl-1-Propanol), supplemented with 9 mM of the ALP substrate, p nitrophenyl-phosphate (pNPP). The cell lysate was diluted 10X in 0.1% Triton X-100 and mixed with an equal volume of prepared AMP buffer for a total volume of 200 µL. Color development was read in situ every two minutes for fourteen minutes at 405 nm in an Infinite 200 PRO multimode plate reader (Tecan Group Ltd., Switzerland). Readings were converted to concentration with a standard curve based on 4-Nitrophenol. ALP results are presented as enzyme activity over time, the rate of p-nitrophenol production from the p-nitrophenyl phosphate substrate (pNPP, see Materials above), and normalized to cell numbers as calculated from DNA content, using PICO green (see above). The normalized results are expressed as the amount of substrate converted (ng) over time per number of cells (ng/min/10k cells).

### 4.7. Alizarin Red and Dissolved Calcium TECO Assay for Mineralization

After 21 days in culture, mineralization in the cultures was assessed qualitatively by Alizarin Red staining, essentially as previously described [35,67]. In brief, following washing with PBS (at least three times) and fixation with 10% neutral buffered formalin for 15min, the cultures were stained using 0.5% Alizarin Red S (pH 4.2) for visual confirmation of mineralization. Mineralization was quantified using a commercially available, colorimetric calcium quantification kit (Teco Diagnostics, Anaheim CA), following destructive decalcification of the cultures in 0.6 N HCl and analyzing the supernatant according to the manufacturer’s instructions, as previously described [35]. Separate standard curves were established for each experiment.

### 4.8. RNA Extraction and Real-Time PCR for Osteogenic Marker Gene Expression (ALPL, RUN, ON)

The expression of select osteogenic marker genes was assessed by quantitative PCR (qPCR), essentially as previously described, with some minor modifications [35,67]. The Taqman primers (from Allied Biosystems, ThermoFisher) were: ALPL (Mm00475834_m1), RUNx2 (Mm00501584_m1) and Osteonectin/SPARC/BM40 (Mm00486332_m1). Quantitative PCR (qPCR) was performed in a RealPlex Real-Time PCR System (Eppendorf, Enfield, CT) with fast thermal cycling, as previously described [35]. The level of expression of each gene was normalized to the level of expression of a common standard housekeeping gene, glyceraldehyde 3-phosphate dehydrogenase (GAPDH), and a relative expression set against the control at 1G in MM. Fold change was calculated via the comparative CT method (2^−ΔΔCT^) [68].

### 4.9. Statistical Analysis

Statistical differences between the samples were assessed by ANOVA and post-hoc analysis using Tukey’s HSD (honestly significant difference) test unless otherwise specified. Mean Absolute Error (MAE) was used to determine the difference between modeled projections. Results were plotted using Excel or JMP Pro. Data are presented as means ± standard deviation from at least three independent experiments (*n* = 3 biological replicates). *p* < 0.05 was considered significant and noted as “*”, *p* < 0.01, and *p* < 0.001 were noted as “**” and “***”, respectively.

## 5. Conclusions

In this study, we used the osteoblastic cell line 7F2 to test the hypothesis that several antioxidant nutraceuticals might promote osteoblast differentiation and function and thus potentially mitigate inhibition due to reduced gravity. We demonstrated that the phytochemical derivatives curcumin and carnosic acid and the trace element zinc can promote osteoblastic cell growth in the absence of traditional osteogenic media. All three nutraceuticals stimulated osteogenic differentiation, as assessed by elevated enzymatic ALP activity and changes in the expression of osteogenic marker genes. The osteogenic effects of the plant-derived nutraceuticals were identified as synergistic. Cells treated with the nutraceuticals, especially with the synergistic intermix, were able to counteract simulated microgravity-based inhibition primarily through heightened levels of tissue-nonspecific alkaline phosphatase. We surmise that intermixes of phytonutrients may serve as convenient and effective nutritional countermeasures against bone loss in space.

## Figures and Tables

**Figure 1 ijms-23-00136-f001:**
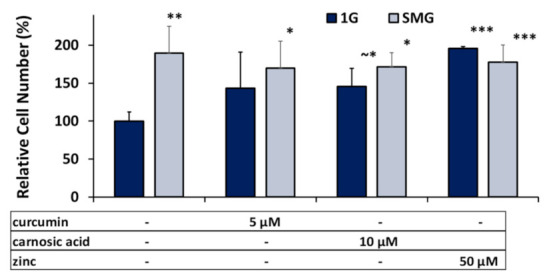
Effects of optimal nutraceutical concentrations on relative cell density per flask. Cell numbers after 6 days’ DM cell culture in 1G or SMG were quantified using the PICO green fluorometric assay and normalized to the 1G controls. Error bars indicate standard deviations. (~*) *p* ≈ 0.05; (*) *p* < 0.05; (**) *p* < 0.01; (***) *p* < 0.001. The results reflect means ±SD from three independent experiments (*n* = 3 biological replicates).

**Figure 2 ijms-23-00136-f002:**
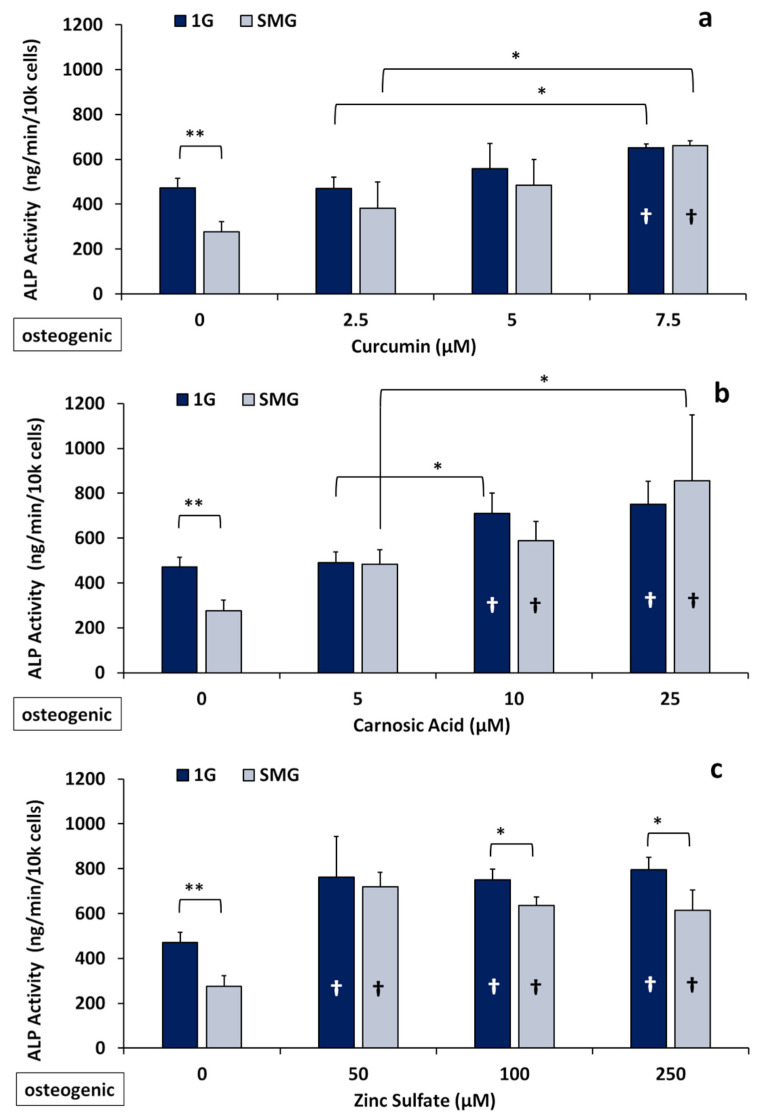
Effects of nutraceuticals on ALP activity of 7F2 cells after 6 days of culture (*n* = 3) in 1G and SMG in osteogenic medium. Confluent cell cultures were treated with different concentrations of the nutraceuticals and ALP activity was assessed as described in Materials and Methods. (**a**) Curcumin; (**b**) carnosic acid; and (**c**) zinc sulfate. (*) *p* < 0.05; (**) *p* < 0.01. † (inside the bars) indicates a significant difference from control. The results reflect means ±SD from three independent experiments (*n* = 3 biological replicates). Statistical analysis was conducted using ANOVA followed by Tukey’s post-hoc test.

**Figure 3 ijms-23-00136-f003:**
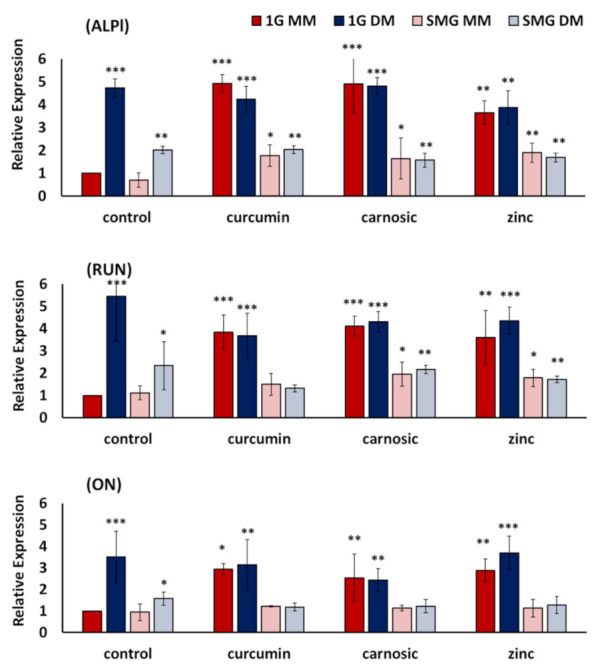
Effects of nutraceuticals on the expression of osteogenic marker genes ALPl, RUN, and ON. 7F2 cells were cultured for six days in 1G and SMG, respectively, in non-osteogenic maintenance media (MM) and osteogenic differentiation media (DM). Gene expression was analyzed by real-time PCR and normalized to the levels of a housekeeping gene (GAPDH), with 1G MM as control. The expression of each gene to internal control is presented as fold-change expression for each transcript, with control as 1-fold expression. For details, see Materials and Methods. Nutraceutical concentrations were 7.5 µM curcumin, 10 µM carnosic acid, 50 µM zinc. Values are Means +/− SD of *n* = 3. Versus 1G-MM-Control, Asterisk (*) *p* < 0.05, (**) *p* < 0.01, (***) *p* < 0.001. The results reflect means ± SD from three independent experiments (*n* = 3 biological replicates).

**Figure 4 ijms-23-00136-f004:**
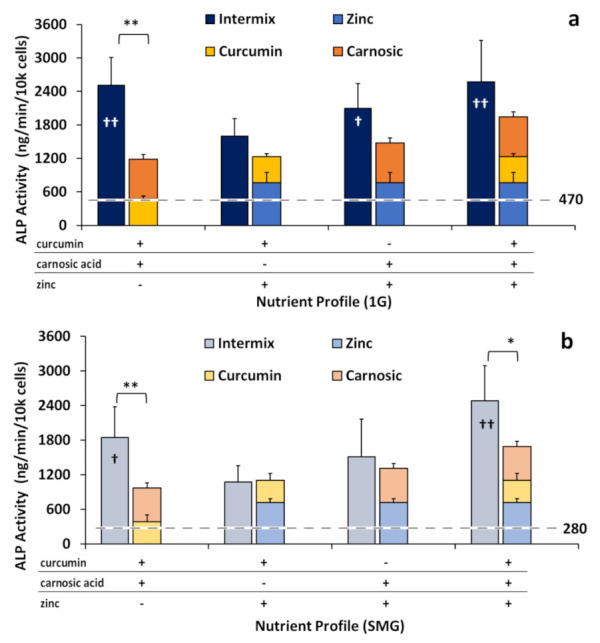
ALP activity normalized by cell number, with ALP activity hypothetically stacked additively versus observed in intermixes. Intermixes are 3.75 µM curcumin + 12.5 µM carnosic acid, 3.75 µM curcumin + 25 µM zinc, 12.5 µM carnosic acid + 25 µM zinc, and 3.75 µM curcumin + 12.5 µM carnosic Acid + 25 µM zinc. (**a**) 1G; (**b**) SMG. The dotted lines represent the levels of normal ALP activity in either 1G (**a**) or SMG (**b**). Asterisk (*) *p* < 0.05, (**) *p* < 0.01; Cross (†) *p* < 0.05, (††) *p* < 0.01, versus 1G controls. The results reflect means ±SD from three independent experiments (*n* = 3 biological replicates).

**Figure 5 ijms-23-00136-f005:**
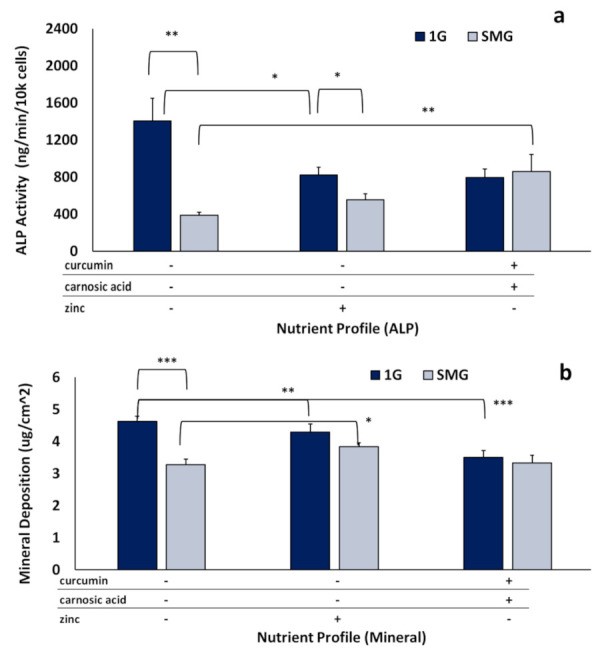
Effects of nutraceuticals on long-term ALP activity and mineral deposition. Long-term SMG experiments were conducted over 21 days in DM to assess enzymatic ALP and mineral deposition using “optimal” nutraceutical mixes: (“phytonutrients”) 3.75 µM curcumin + 12.5 µM carnosic acid and (“zinc”) 25 µM zinc. (**a**) ALP activity normalized by cell number. (**b**) mineralization quantified as micrograms of calcium per square centimeter (µg/cm^2^). Asterisk (*) *p* < 0.05, (**) *p* < 0.01, (***) *p* < 0.001. The results reflect means ±SD from three independent experiments (*n* = 3 biological replicates).

**Table 1 ijms-23-00136-t001:** Effects of nutraceutical concentrations on relative cell viability. After the initial 6-h exposure, viability relative to untreated control. Data depicted as average ± standard deviation. sterisk (*) *p* < 0.05, (**) *p* < 0.01, (***) *p* < 0.001 for a given concentration and nutraceutical.

Curcumin		Carnosic Acid		Zinc	
2.5 µM	89.7 ± 2.4 *	5 µM	85.1 ± 1.2	50 µM	88.2 ± 3.5
5 µM	71.6 ± 2.5 ***	10 µM	82.3 ± 2.1	100 µM	79.7 ± 2.5 *
7.5 µM	65.3 ± 4.5 ***	25 µM	65.8 ± 0.8 *	250 µM	71.7 ± 8.3 *
15 µM	42.9 ± 1.5 ***	50 µM	48.9 ± 13.2 **	500 µM	67.3 ± 9.3 *

## Data Availability

The data of this study are available from the authors upon reasonable request.

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
