# Peer review of "Nutraceuticals Synergistically Promote Osteogenesis in Cultured 7F2 Osteoblasts and Mitigate Inhibition of Differentiation and Maturation in Simulated Microgravity"

_ijms, 2021, doi:10.3390/ijms23010136_

Round 1

Reviewer 1 Report

Dear Authors,

I commend you on this remarkable manuscript. As stated, nutraceuticals are studied broadly these days in inflammatory diseases and cancer, but may have prominent implications in bone health as well. To test curcumin, carnosic acid, and zinc in a simulated microgravity environment is an elegant method to try to tackle the problems at hand not only for space travelers, as well as supplement research on other issues of bone health as osteoporosis.

Beside a few punctuation errors and formatting issues (figures 2.b, 4.a, 4.b, 5.a, 5.b,) I cannot find any lacking parts or necessary changes. 

In a follow up manuscript would be interested to see the pathways of the reported changes in osteogenesis elucidated. As the authors suggested, miRNAs might be have a leading role in this, maybe communicated by EVs.

Author Response

We are grateful to reviewer 1 for his/her kind words.

  1. We have rechecked the manuscript (using Grammarly) and hopefully eliminated any remaining punctuation/typographical errors
  2. We apologize for the formatting issues in Figures 2b, 4a, 4b, 5a, and 5b (technical glitches due to overlapping text inserts in Word). We have eliminated these formatting errors and hope the figures are now publication quality.
  3. We have briefly addressed the well-taken suggestion regarding future mechanistic studies. The last sentence in the first paragraph on p12 (discussion) reads now: Going forward, we suggest focusing on the mechanisms by which the three nutraceuticals studied can improve bone health in microgravity, for example by altering post-translational regulation of the transcriptional activity of osteogenic genes, or serving as modulators of the epigenetics of osteogenic miRNAs.

Reviewer 2 Report

The authors convincingly describe the influence of curcumin, carnosic acid, and zinc on cell density, ALP activity, and expression of the osteogenic marker genes ALPI, RUN and ON under 1G and simulated microgravity. The study has relevance as the suggestion of an intermix of phytonutrients potentially effects the health care in space.

The conclusions are plausible, but two formulations mislead the reader.

  1. Figure 1, Description of significant differences: It would be more intuitive if the 1G control was specified in more detail (1G without nutraceutics for example).
  2. Results page 6/7: “For example, ALPl expression in non-osteogenic maintenance media was elevated to levels similar to the levels seen in osteogenic media.” The description is not comprehensible because the reference point is unclear.

A major criticism is that the authors do not specify the number of replicates in the experiments. This should be corrected as otherwise the value of P-values and standard deviations cannot be estimated. Furthermore, the specification of the used rtPCR primers would be helpful to simplify the replication of the study.

Minor point: On results page 6 GAPH instead of GAPDH is written.

Author Response

We thank reviewer 2 for his/her constructive criticisms. We apologize for the potentially ambiguous formulations and have attempted to clarify the issues as follows:

  1. In describing Figure 1 we clarified the significant differences according to the reviewer’s suggestions. The sentence in the legend to Figure 1 (page 4) now reads: Significant differences (asterisks) above columns display deviation from 1G control, in the absence of the nutraceuticals. In addition, we also made some minor modifications, which will hopefully further clarify the text describing Figure 1 (page 3 bottom).
  2. In response to the reviewer’s well-taken criticism of the description (on page 6/7) of the results depicted in Figure 3, we have revised the text to read: Surprisingly, while the nutraceuticals did not significantly alter gene expression in osteogenic differentiation media, the nutraceuticals did significantly induce the expression of all osteogenic genes in non-osteogenic maintenance media (Figure 3). As an example, in SMG, in the presence of all nutraceuticals, ALPl expression in non-osteogenic maintenance media (SMG MM) was elevated to levels statistically similar to that seen in SMG osteogenic media (SMG DM). This contrasts with osteogenic differentiation media (DM), where the nutraceuticals did not change the expression of ALPI, Runx2, or ON. While the nutraceuticals did not effectively counteract the inhibition of osteogenic marker gene expression in SMG, supplementation with the nutraceuticals raised the expression levels of 1G MM to 1G DM control and SMG MM to SMG DM control. Using ALPI as an example, the increases observed for cells cultured in MM-media were 160% for curcumin, 130% for carnosic acid, 170% for zinc, respectively.
  3. We apologize for the oversight of not having listed the number of replicates for the data in each figure. In the revised version, we clarified this in general in the section on statistics in Materials and Methods (p 15. Data are presented as means ± standard deviation from at least three independent experiments (n=3 biological replicates) as well as in the legends for each Figure separately( The results reflect means ± D from three independent experiments (n=3 biological replicates).
  4. For the qualitative PCR studies, we did not use conventional PCR primers, instead we, like most others employing qPCR, used the validated Taqman primers, which are described in section 4.8 of the Materials and Methods section ( p14).